

# Community Intercomparison Suite (CIS) v1.3.2: A tool for intercomparing models and observations

Duncan Watson-Parris[1,2], Nick Schutgens[2], Nicholas Cook[1], Zak Kipling[2], Philip Kershaw[3,4], Edward Gryspeerdt[5], Bryan Lawrence[3,6,7], and Philip Stier[2]

[1]Tessella Ltd, Abingdon, Oxford, OX14 3YS
[2]Atmospheric, Oceanic and Planetary Physics, Department of Physics, University of Oxford, Oxford, UK
[3]Centre for Environmental Data Analysis, STFC Rutherford Appleton Laboratory, Didcot, United Kingdom
[4]National Centre for Earth Observation, United Kingdom
[5]Institute for Meteorology, Universität Leipzig, Leipzig, Germany
[6]Department of Meteorology, University of Reading, Reading, United Kingdom
[7]National Centre for Atmospheric Science, United Kingdom

*Correspondence to:* Duncan Watson-Parris (duncan.watson-parris@physics.ox.ac.uk)

**Abstract.** The Community Intercomparison Suite (CIS) is an easy-to-use command-line tool which has been developed to allow the straightforward intercomparison of remote sensing, in-situ and model data. While there are a number of tools available for working with climate model data, the large diversity of sources (and formats) of remote sensing and in-situ measurements necessitated a novel software solution. Developed by a professional software company, CIS supports a large number of gridded and ungridded data sources 'out-of-the-box', including climate model output in NetCDF or the UK Met Office pp file format, CALIOP (Cloud-Aerosol Lidar with Orthogonal Polarization), MODIS (MODerate resolution Imaging Spectroradiometer), Cloud and Aerosol CCI (Climate Change Initiative) level 2 satellite data, and a number of in-situ aircraft and ground station datasets. The open-source architecture also supports user defined 'plugins' to allow many other sources to be easily added. Many of the key operations required when comparing heterogenous datasets are provided by CIS, including subsetting, aggregating, collocating and plotting the data. Output data is written to CF-compliant NetCDF files to ensure interoperability with other tools and systems. The latest documentation, including a user manual and installation instructions can be found on our website (http://cistools.net). Here we describe the need which this tool fulfils, followed by descriptions of its main functionality (as at version 1.3.2) and plugin architecture which make it unique in the field.

## 1 Introduction

Modern Global Climate Models (GCMs) produce huge amounts of prognostic and diagnostic data covering every aspect of the system being modelled. The upcoming CMIP-6 (Coupled Model Intercomparison Project Phase 6) is likely to produce more than 50Pb of data alone. Analysis of the data from these models forms the cornerstone of the IPCC (Stocker et al., 2013) (Intergovernmental Panel on Climate Change) and subsequent UNFCCC (United Nations Framework Convention on Climate Change) reports on anthropogenic climate change, but there exist large differences across the models in a number of key climate variables (e.g. Boucher et al., 2013; Suzuki et al., 2011). In order to understand these differences and improve the models the



model data must be compared not only with each other – which is relatively straightforward for large intercomparison projects such as CMIP which provide the model data in a common data standard, but also with observational data – which can be much harder.

Observational data can also be extremely voluminous. For example modern Earth Observation (EO) satellites can easily produce petabytes of data over their lifetime. There are dozens of EO satellites being operated by the National Aeronautics and Space Administration (NASA), the European Space Agency (ESA), and other international space agencies. While modern missions use common data standards there are many valuable datasets stored in unique formats and structures which were designed when storage was at more of a premium, and so are not particularly user-friendly. Ground based EO sites, and in-situ measurement of atmospheric properties are also areas where many different groups and organisations produce data in a wide variety of formats.

The process of model evaluation typically involves a relatively small set of common operations on the data: reading, subsetting, aggregating, analysis and plotting. Many of these operations are currently written as a bespoke analysis for each type of data being compared. This is time consuming and error prone. While a number of tools currently support the comparison and analysis of model data in standard formats, such as NetCDF Operators (NCO) (Zender, 2008), Climate Data Operators (CDO) (http://www.mpimet.mpg.de/cdo), Iris (Met Office, 2015), and cf-python (http://cfpython.bitbucket.org) there are few if any which support observational data. A tool described by Langerock et al. (2015) provides some of this functionality for a specific set of observational data. There are also some websites which allow a pre-defined analysis of specific datasets (for example Giovani: http://giovanni.sci.gsfc.nasa.gov), but do not give the flexibility of a tool which can be installed and run locally. The Community Intercomparison Suite (CIS) seeks to fill this gap: the primary goal of CIS is to provide a single, flexible tool for the quantitative and qualitative intercomparison of remote-sensing, in-situ and model data.

Further, primarily because of this lack of appropriate tools, many intercomparisons and evaluations are carried out using aggregated data (see Appendix A for a definition of 'aggregated data', and other terms). While gridded observational data is often available (e.g. Teixeira et al., 2014), and can be much easier to work with, considerable error can be introduced during the aggregation, particularly for sparse datasets (Levy et al., 2009). Further, point-wise comparison can be shown to provide improved confidence when constraining aerosol processes in climate models compared to an aggregated comparison (e.g. Kipling et al., 2013; Schutgens et al., 2016). However, this point-wise comparison requires the collocation of the datasets on to a common spatio-temporal sampling, which can be a complex operation. By providing collocation tools for both gridded and ungridded data CIS allows straightforward point-wise comparison between all of the supported data formats.

In this paper we first describe the development of this new tool (Sect. 2) and the architecture designed to allow maximum flexibility in the data types and functionality supported (Sect. 3). Then tables of the specific types of data CIS supports and detailed descriptions of the operations which can be performed on them are provided (Sect. 4), followed by an example of the scientific workflow which this tool enables (Sect. 5). Information about downloading and installing CIS, or accessing the source code, can be found in Sect. 7. A brief description of the plugin architecture and the steps needed for a user to create their own plugins are provided in Appendix B, and a table of definitions in Appendix A. A reference card providing a one-page summary of the various CIS commands is also available as a supplement to this paper.



## 2 Development

CIS has been developed by a professional software development consultancy (Tessella Ltd.) working closely with the Centre for Environmental Data Analysis (CEDA) and the Department of Physics at the University of Oxford to ensure a high quality tool which meets the need of a broad range of users. The use of modern development practices such as Test Driven Development (TDD) (Beck, 2003) and Continuous Integration (CI) (Beck, 2000) has ensured that each component is automatically tested against hundreds of Unit Tests before it is deployed. These test each individual function within the code to ensure defects are kept to a minimum, and particularly reduce regressions (defects introduced into code which was previously working)

The development was also carried out in an Agile fashion, specifically using Scrum (Schwaber and Beedle, 2001). In this approach regular working releases were made at the end of two-week implementation 'sprints', each delivering a prioritised set of fully functioning requirements adding immediate value to the users (scientists). The developers were supported by a Subject Matter Expert (SME) who worked with the scientists to define and prioritise each area of development (User Stories); a dedicated testing specialist who was responsible for defining and performing independent testing; and a Project Manager (PM) who oversaw progress and managed the overall development process from the Tessella perspective.

CIS is completely written in Python, which provides a good balance between speed, versatility, and maintainability, and allows easy installation across many platforms (see Sect. 7 for more details). Python also has many open-source libraries available to build on, and in particular CIS makes heavy use of the Iris (Met Office, 2015) library for it's internal representation of gridded data types. Many of the more numerically intensive operations within CIS are performed by Python libraries with interfaces to C implementations to keep the run-time as fast as possible.

Much consideration was given to the need for parallelisation and optimisation of the functions within CIS, particularly around collocation where long runtimes for large datasets can be expected. Significant development time was devoted to optimisations in these functions and many of the run-times now scale very well with size of the data. However, we deemed it a lower priority to devote development time to parallelising these operations, as they are usually trivially parallelised by the user by performing the operation on each input file separately across the available compute nodes (using a batch script for example, and subsetting the data first as needed). Such a script is pre-installed alongside CIS on the UK JASMIN big-data analysis cluster (Lawrence et al., 2012) and could be easily ported to other clusters.

All of the source code for CIS is freely available under the GNU Lesser General Public License v3, which it is hoped will promote widespread uptake of the tool, and also encourage wider collaboration in its development.

## 3 Extensible architecture

One of the key features of CIS is the flexible and extensible architecture. From the outset it was obvious that there was no way for a single, unextendable, tool to provide compatibility with the wide variety of data sources available and support all of the various analyses which would be performed on them. A modular design was therefore incorporated which allowed user-defined components to be swapped in as easily as possible.



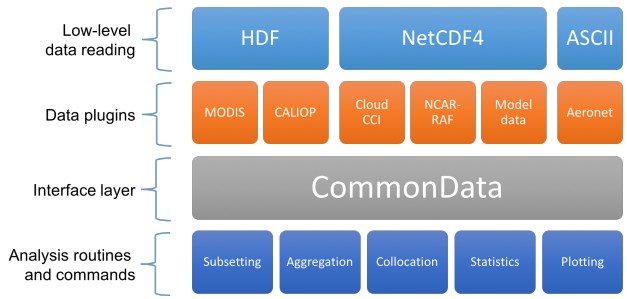

**Figure 1.** An illustration of the architecture of CIS demonstrating the different components in the modular design.

At the heart of the design is the 'CommonData' interface layer which allows each of the analysis routines and commands to work independently of the actual data being provided, as shown in Fig. 1. The top row of modules in this figure represent the low-level reading routines which are used for actually reading and writing data to the different data formats. The orange components are the data products which interpret the data for CIS and can be swapped out by the user (using 'plugins', as

described in Sect. B1). The CommonData block represents the internal CIS data structure which abstracts the CIS functionality (shown in the bottom row) from the different data formats above. Specifically, CommonData is an abstract base class (a class defines an object in object oriented programming) which defines a number of methods which the analysis routines can assume will exist regardless of the underlying data. The two main concrete types of CommonData are GriddedData and UngriddedData, which represent gridded and ungridded data respectively.

There are an extensive number of data sources which are supported by CIS, which can be broadly categorised as either 'gridded' or 'ungridded' data. Gridded data is defined as any regularly gridded data-set for which points can be indexed using $(i, j, k, ...)$ where $i$, $j$ and $k$ are integer indexes on a set of orthogonal coordinates (see Fig. 2a). Here we define the gridded data values as an $n$-dimensional matrix $\mathbf{G}$ and $n$ coordinate vectors $\mathbf{x}, \mathbf{y}, ...$, which we will use in the algorithmic descriptions of the operations in Sect. 4. Ungridded data is anything which does not meet this criteria, and in general it is assumed each $(x, y, z)$

point is independent of every other point (Fig. 2b). Then we can define a data value $u_j$ and set of coordinates $\mathbf{r}_j$ at each point $j$. Note that although this independence may not strictly be true for some of the data sources (for example satellite mounted LIDAR instruments where many altitude points will be present for each latitude/longitude point) this strict definition applies within CIS. This allows significant optimisations to be made for operations on gridded data, and flexibility in dealing with ungridded data, at the expense of performance for some operations on those ungridded datasets which do have some structure.

In CIS the gridded data type is really just a thin wrapper around the 'cube' provided by the Iris (Met Office, 2015) library. All of the ungridded routines are however bespoke and include a number of useful features (besides the main analysis routines) including multi-file and multi-variable operations, hierarchical NetCDF file reading and automatic recognition of file-types. The ungridded data is stored internally as one numpy (van der Walt et al., 2011) array of values and a set of associated metadata. There is one such structure for the data values themselves, and each of the latitude, longitude, time and altitude coordinates (as

needed). These arrays may take on any shape, though they must all be the same.



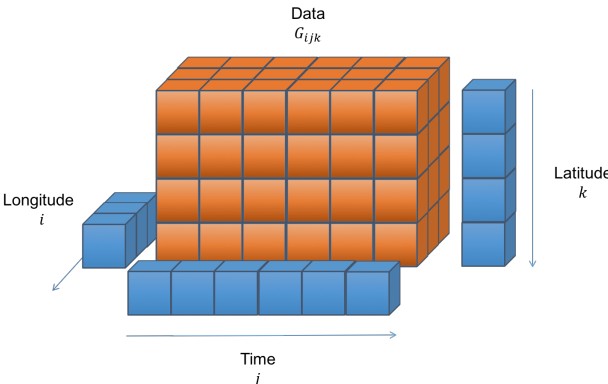

**Figure 2a.** An illustration of the design of the *gridded* data objects used internally by CIS – based heavily on the Iris 'cube'. The $n$-dimensional data array **G** is accompanied by $n$ 1-dimensional coordinate arrays. Note that hybrid height and pressure coordinates can also be created and used as needed.

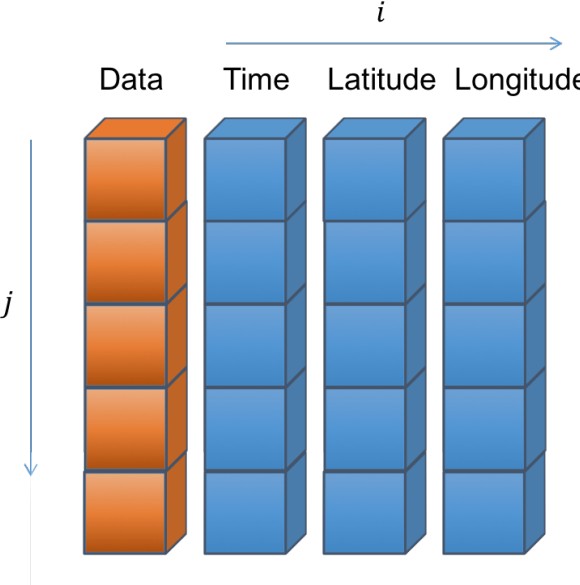

**Figure 2b.** An illustration of the design of *ungridded* data objects used internally by CIS. All $j$ points are assumed to be independent of each other. The data and associated coordinates are represented as a series of 1-dimensional arrays.

## 4    Core functionality

In this section we describe the core functionality of CIS. Each sub-section gives a brief description of an operation; the command line syntax and expected output; a formal algorithmic description of the operation (where appropriate); and a short example.



**Table 1.** A list of the ungridded data sources supported by CIS 1.3.2. The file signature is used by CIS to automatically determine the correct product to use for reading a particular set of data files, although this can easily be overridden by the user. (Internally these signatures are represented as Python regular expressions, here they are shown a standard wildcards for ease of reading.)

| Dataset | Product name | Type | File Signature |
|---|---|---|---|
| MODIS L2 | MODIS_L2 | Satellite | *MYD06_L2*.hdf, *MOD06_L2*.hdf, *MYD04_L2*.hdf, *MOD04_L2*.hdf, *MYDATML2.*.hdf, *MODATML2.*.hdf |
| Aerosol CCI | Aerosol_CCI | Satellite | *ESACCI*AEROSOL* |
| Cloud CCI | Cloud_CCI | Satellite | *ESACCI*CLOUD* |
| CALIOP L1 | Caliop_L1 | Satellite | CAL_LID_L1-ValStage1-V3*.hdf |
| CALIOP L2 | Caliop_L2 | Satellite | CAL_LID_L2_05kmAPro-Prov-V3*.hdf |
| CloudSat | CloudSat | Satellite | *_CS_*GRANULE*.hdf |
| NCAR-RAF | NCAR_NetCDF_RAF | Aircraft | *.nc containing the attribute Conventions with the value NCAR-RAF/nimbus |
| GASSP | NCAR_NetCDF_RAF | Aircraft | *.nc containing the attribute GASSP_Version |
| GASSP | NCAR_NetCDF_RAF | Ship | *.nc containing the attribute GASSP_Version, with no altitude |
| GASSP | NCAR_NetCDF_RAF | Ground-station | *.nc containing the attribute GASSP_Version, with attributes Station_Lat, Station_Lon and Station_Altitude |
| AERONET | Aeronet | Ground-stations | *.lev20 |
| CSV datapoints | ASCII_Hyperpoints | N/A | *.txt |
| CIS ungridded | cis | CIS output | cis-*.nc |

In order to keep the formal algorithmic descriptions concise without any loss of accuracy we adopt a mixture of set and vector notation, and define that notation here. It is useful to define vector inequalities as:

$$\mathbf{x} \leqq \mathbf{y} \text{ if } x_i \leq y_i \text{ for } i = 1, \dots, n \tag{1}$$

Similar identities can be defined for the other inequalities. We use $\emptyset$ to denote an empty set, $\forall$ should be read as 'for all', : as 'such that', and $\lor$ as logical 'and'. Some operations involve the use of a kernel to reduce a set to a single value, we denote these as $\mathcal{K}$.

Although data reading is something a user is rarely aware of when using CIS, the flexibility offered in this regard is an important distinguishing feature. All of the functions described in the following sections are possible with any of the supported datasets, and any datasets supported by user written plugins (as described in Sect. B1).

A list of the ungridded data sources supported by CIS out-of-the-box is presented in Table 1, and gridded data sources in Table 2. As CIS uses Iris for gridded data support, any Climate and Forecast (CF) compliant (http://cfconventions.org/Data/cf-conventions/cf-conventions-1.6/build/cf-conventions.pdf) NetCDF4 data can be read in with CIS, as well as the other formats listed.



**Table 2.** A list of the gridded data sources supported by CIS 1.3.2. The file signature is used by CIS to automatically determine the correct product to use for reading a particular set of data files. This can always be overridden by the user.

| Dataset | Product name | Type | File Signature |
|---|---|---|---|
| Net_CDF Gridded Data | NetCDF_Gridded | Any CF-NetCDF4 compliant gridded data | *.nc (this is the default for NetCDF Files that do not match any other signature) |
| MODIS L3 daily and 8-day | MODIS_L3 | Satellite | *MYD08_D3*.hdf, *MOD08_D3*.hdf, *MOD08_E3*.hdf |
| UK Met Office pp data | HadGEM_PP | Gridded Model Data | *.pp |

### 4.1 Subsetting

Subsetting allows the reduction of data by extracting variables and restricting them to user specified ranges in one or more coordinates. Both gridded and ungridded datasets can be reduced in size by specifying the range over which the output data should be included, and points outside that range are removed.

5    The basic structure of the subset command is:

```
$ cis subset <datagroup> <limits> [-o output_file]
```

Where 'subset' is the sub-command to invoke in 'cis' and the 'output_file' is the (optional) filename to be used for outputting the result. If none is specified then a default is used. The two main arguments 'datagroup' and 'limits' are more complex and will be discussed below.

10    The 'datagroup' is a common concept across the various CIS commands. It represents a collection of variables (from a collection of files) sharing the same spatio-temporal coordinates, which takes the form:

```
variables:filenames[:product=...]
```

Here, the 'variables' element specifies the variables to be operated on and can be a single variable, a comma separated list, a wildcarded variable name, or any combination thereof. The 'filenames' element specifies the files to read the variables from and can be a single filename, a directory of files to read, a comma separated list of files or directories, wildcarded filenames, or any combination thereof. The optional 'product' element can be used to manually specify the particular product to use for reading this collection of data. See Tables 1 and 2 for a full list of initially available product names.

The 'limits' are a comma separated list of the upper and lower bounds to be applied to specific dimensions of the data. The dimensions may be identified using their variable names (e.g. 'latitude') or by choosing a shorthand from 'x', 'y', 'z', 'p', or

20    't' which refer to longitude, latitude, altitude, pressure and time respectively. The limits are then defined simply using square brackets, e.g. x=[-10,10]. The use of square brackets is a useful reminder that the intervals are inclusive, as discussed below. A time dimension can be specified as an explicit window as `t=[2010-01-01T00,2010-12-31T23:59:59]`, or more simply as a single value: `t=[2010]`. In this case the value is interpreted as both the start and the end of the window and all points which fall within 2010 would be included. For all other dimensions the units used are those from the datafile.





The detailed algorithm used for subsetting ungridded data is outlined in Algorithm 1, and for gridded data in Algorithm 2. The algorithms use a mix of pseudo-code and mathematical notation to try to present the operations in a clear, but accurate way. The operations themselves will involve other checks and optimisations not shown in the algorithms, but the code is available for those interested in its exact workings. See Sect. 3 for definitions of the gridded and ungridded entities.

---

**Algorithm 1** Subset an ungridded dataset given lower and upper bounds ($\mathbf{a}$ and $\mathbf{b}$ respectively) and return the subset as $o$.

---

INITIALIZE $o = \emptyset$

**for** j=1,...,J **do**

$o_j = u_j$ iff $\mathbf{a} \leqq \mathbf{r_j} \leqq \mathbf{b}$

**end for**

---

**Algorithm 2** Subsetting a gridded dataset $\mathbf{G}$. Here we define the algorithm for a two-dimensional dataset, although it can trivially be extended to higher dimensions. Given lower and upper bounds, $a$ and $b$ respectively for each dimension, return the subset as $\mathbf{O}$.

---

**if** cells have upper and lower bounds $x^u$ and $x^l$ respectively **then**

$\mathbf{O} = \{G_{ij} \forall i,j : (a_x \leq x_i^u \vee x_i^l \leq b_x) \vee (a_y \leq y_j^u \vee y_j^l \leq b_y)\}$

**else**

$\mathbf{O} = \{G_{ij} \forall i,j : (a_x \leq x_i \leq b_x) \vee (a_y \leq y_j \leq b_y)\}$

**end if**

---

5  For example the following command would take the variable 'aod550' from the file '`satellite_data.hdf`' and output the data contained in a lat/lon region around North America for the 4th of February 2010 to a file called '`subset_of_satelite_data.nc`':

```
$ cis subset aod550:satellite_data.hdf x=[-170,-60],y=[30,85],t=[2010-02-04] -o
    subset_of_satelite_data
```

10  The output file is stored as a CF compliant NetCDF4 file.

### 4.2 Aggregation

CIS also has the ability to aggregate both gridded and ungridded data along one or more coordinates. For example, it can aggregate a model dataset over the longitude coordinate to produce a zonal mean, or aggregate satellite imager data onto a 5° lat/lon grid.

15  The aggregation command has the following syntax:

```
$ cis aggregate <datagroup>[:options] <grid> [-o outputfile]
```




Where 'aggregate' is the sub-command; 'datagroup' specifies the variables and files to be aggregated (see Sect. 4.1 for more details); the 'options' define a number of choices available for fine tuning the aggregation which are detailed below; and 'grid' defines the grid which the data should be aggregated onto.

The optional arguments should be given as `keyword=value` pairs in a comma separated list. The only currently available option (other than the 'product' option described in the datagroup summary above) is the 'kernel' option. This allows the user to specify the exact aggregation kernel to use. If not specified the default is 'moments' which returns the number of points in each grid cell, their mean value, and the standard deviation in that mean. Other options include 'max' and 'min' which return the maximum and minimum value in each grid cell respectively.

The mandatory 'grid' argument specifies the coordinates to aggregate over. The detail of this argument, and the internal algorithms applied in each case are quite different when dealing with gridded and ungridded data so they will be described separately below. This difference arises primarily because gridded data can be completely averaged over one or more dimensions, and also often requires area weights to be taken into account.

### 4.2.1 Ungridded aggregation

In the case of the aggregation of ungridded data the mandatory 'grid' argument specifies the structure of the 'binning' to be performed for each coordinate. The user can specify the start, end and step size of those bins in the form `coordinate=[start, end,step]`. The 'step' may be missed out, in which case the bin will span the whole range given. Coordinates may be identified using their variable names (e.g. 'latitude') or by choosing from 'x', 'y', 't', 'z', 'p' which refer to longitude, latitude, time, altitude and pressure respectively. Multiple coordinates can be aggregated over, in which case they should be separated by commas.

The output of an aggregation is always regularly gridded data, so CIS does not currently support the aggregation over only some coordinates. If a coordinate is not specified (or is specified, but without a step size) then that coordinate is completely collapsed. That is, we average over its whole range, so that the data is no longer a function of that coordinate. Specifically, one of the coordinates of the gridded output would have a length of one, with bounds reflecting the maximum and minimum values of the collapsed coordinate.

The algorithm used for the aggregation of ungridded data is identical to ungridded to gridded collocation (as this is essentially an collocation operation with the grid defined by the user) described in Algorithm 4.

An example of the aggregation of some satellite data which contains latitude, longitude and time coordinates is shown below. In this case we explicitly provide a one degree by one degree latitude and longitude grid, and implicitly average over all time values.

```
$ cis aggregate AOT500:satellite_data.nc:kernel=mean x=[-180,180,1],y=[-90,90,1] -o
    agg-out.nc
```



### 4.2.2 Gridded aggregation

For gridded data the binning described above is not currently available, this is partly because there are cases where it is not clear how to apply area weighting. The user is able to perform a complete collapse of any coordinate however, simply by providing the name of the coordinate(s) as a comma separated list – e.g. 'x,y' will aggregate data completely over both latitude and longitude, but not any other coordinates present in the file.

The algorithm used for this collapse of gridded dimensions is more straightforward than that of the ungridded case. First the area weights for each cell are calculated, and then the dimensions to be operated on are averaged over simultaneously. That is, the different moments of the data in all collapsed dimensions are calculated together, rather than independently (Using the Iris routines described here: http://scitools.org.uk/iris/docs/latest/iris/iris/cube.html#iris.cube.Cube.collapsed), as values such as the standard deviation are non-commuting.

A full example of gridded aggregation, taking the time and zonal average of total precipitation from the HadGEM3 (Hewitt et al., 2011) GCM is shown below. A plot of the resulting data is shown in Fig. 3.

```
$ cis aggregate rsutcs:model_data.nc:kernel=mean t,x -o agg-out.nc
```

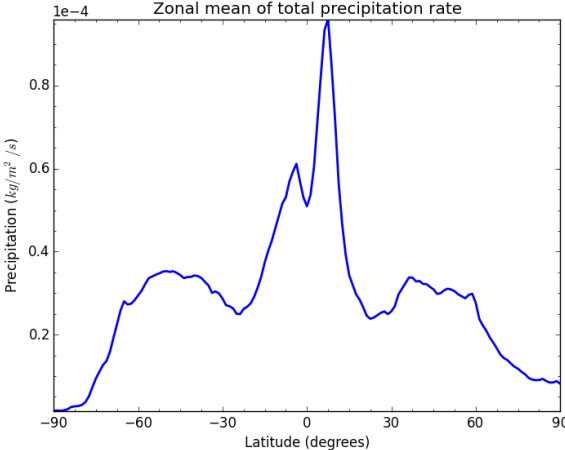

**Figure 3.** A plot of the zonal average of global rainfall, demonstrating the simple aggregation of global model outputs using CIS. See the text for the exact command used to produce this output.

### 4.3 Collocation

Point-wise quantitative inter-comparisons require the data to be mapped onto a common set of co-ordinates before analysis, and CIS provides a number of straightforward ways of doing this. One of the key features of CIS is the ability to collocate one or more arbitrary datasets onto a common set of coordinates, for example collocating aircraft data onto hybrid-sigma model





levels, or satellite data with ground-station data. The options available during collocation depend on the types of data being analysed as demonstrated in Table 3. The points which are being mapped on to are referred to as sample points, and the points which are to be mapped as data points.

**Table 3.** An outline of the permutations of collocations types, as a function of the structure of the data and sampling inputs. The available kernels are described in Table 4. Each collocation algorithm is described in more detail in sections 4.3.1 – 4.3.4.

| Sample / Data | Gridded | Ungridded |
|---|---|---|
| Gridded | **linear interpolation (lin)**, nearest neighbour (nn), box | **nearest neighbour (nn)**, linear interpolation (lin) |
| Ungridded | **bin**, box | **box** |

The basic structure of the collocation command is:

```
5    $ cis col <datagroup> <samplegroup> [-o outputfile]
```

Where the 'datagroup' specifies the data variables and files to read as described above. The 'samplegroup' is analogous to a datagroup, except in this case the data being specified is that of the *sample* data. That is, the points which the data should be collocated onto.

The samplegroup has a slightly different format to the datagroup, as the sample variable is optional, and all of the colloca-
tion options are specified within this construct. It is of the format `filename{:options}`. The 'filename' is one or more filenames containing the points to collocate onto. The available options (which should be specified in a comma separated list) are listed below:

– `variable` is used to specify which variable's coordinates to use for collocation. This is useful if a file contains multiple coordinate systems (common in some model output). Note, that if a variable is specified, missing variable values will not be used as sample points.

– `collocator` is an optional argument that specifies the collocation method. Parameters for the collocator, if any, are placed in square brackets after the collocator name, for example, `collocator=box[h_sep=1km,a_sep=10m]`. If not specified, a default collocator is identified for the data / sample combination. The collocators available, and the one used by default, for each sampling combination of data structures are laid out in Table 3.

– `kernel` is used to specify the kernel to use for collocation methods that create an intermediate set of points for further processing, that is 'box' and 'bin'. The default kernel in both cases is 'moments'. The built-in kernel methods currently available are summarised in Table 4.

A full example would be:



**Table 4.** A list of the different kernels available. Note that not all of the kernels are compatible with all of the collocators.

| Kernel | Description | Compatible collocators |
|---|---|---|
| mean | The arithmetic mean of the sampled values: $\bar{y} = \frac{1}{n}\sum_{i=1}^{n} y_i$ | bin, box |
| stddev | The corrected sample standard deviation of the mean: $\sigma_y = \sqrt{\frac{1}{n-1}\sum_{i=1}^{n}(y_i - \bar{y})^2}$ | bin, box |
| moments | This kernel returns three variables: The number of points in each data sampling; their mean value; and the standard deviation in that mean. | bin, box |
| min | The lowest of all of the sampled values | bin, box |
| max | The highest of all of the sampled values | bin, box |
| nn_horizontal | The value of the data point with the smallest haversine distance from the sample point | box |
| nn_altitude | The value of the data point with the smallest separation in altitude from the sample point | box |
| nn_pressure | The value of the data point with the smallest (relative) separation in pressure from the sample point | box |
| nn_time | The value of the data point with the smallest separation in time from the sample point | box |

```
$ cis col rain:mydata??.* mysamplefile.nc:collocator=box[h_sep=50km,t_sep=6000S],
    kernel=nn_t -o my_col
```

There are also many other options and customisations available. For example, by default all points in the sample dataset are used for the mapping. However (as CIS provides the option of selecting a particular variable as the sampling set) the user is

5  able to disregard all sample points whose values are masked (whose value is equal to the corresponding fill_value). The many different options available for collocation, and each collocator can be found in the user manual (see http://cis.readthedocs.org/en/latest/collocation.html).

In the following sections we describe each mode of collocation in more detail, including algorithmic representations of the operations performed.

10  ### 4.3.1 Gridded to gridded

For a set of gridded data points which are to be mapped on to some other gridded sample the operation is essentially a re-gridding and the user is able to use either: linear interpolation (lin), where the data values at each sample point are linearly interpolated across the cell which the sample point falls; nearest neighbour, for which the data cell nearest to the sample cell can be uniquely chosen in each dimension for every point; and 'box' for which an arbitrary search area can be manually defined

15  for the sampling using Algorithm 3. The interpolations are carried out using the Iris interpolation routines which are described in detail elsewhere (see http://scitools.org.uk/iris/docs/latest/iris/iris/cube.html#iris.cube.Cube.interpolate).

CIS can also collocate gridded datasets with differing dimensions. Where the sample array has dimensions that do not exist in the data, those dimensions are ignored for the purposes of the collocation and will not be present in the output. Where the





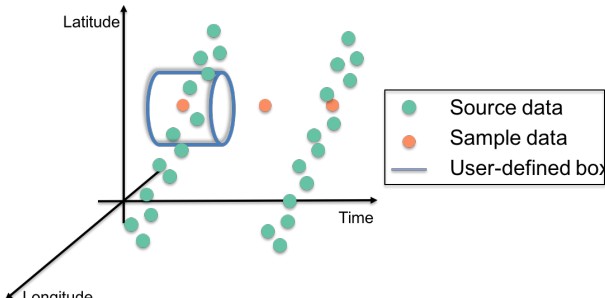

**Figure 4.** This schematic shows the components involved in the collocation of ungridded data onto an ungridded sampling.The user-defined box around each sampling point provides a selection of data points which are passed to the kernel. Note that the resampled data points lie exactly on top of the sample points (which aren't visible).

data has dimensions that do not exist in the sample array, those dimensions are ignored for the purposes of the collocation, and *will* be present in the output.

### 4.3.2 Ungridded to ungridded

CIS is also able to collocate ungridded data. For ungridded to ungridded collocation the user is able to define a 'box' to constrain
5  the data points which should be included for each sample point. The schematic in Fig. 4 shows this box and its relation to the sample and data points. This box can be defined as a distance from the sample point in any of time, pressure, altitude or horizontal (haversine, or great-circle) distance. In general there may be many data points selected in this box. The user also has control over the kernel to be applied to these data values. The default kernel, if none is specified, is the 'moments' kernel which returns the number of points selected, their mean value and the standard deviation on that mean as separate NetCDF variables
10 in the output. Otherwise the user can select only the mean, or the nearest point in either time, altitude, pressure or horizontal distance. In this way the user is able to find, for example, the nearest point in altitude within a set horizontal separation cut-off.

The specific process is outlined in Algorithm 3. For simplicity we have assumed the dimensionality of the datasets is the same, in reality this need not be the case. CIS will collocate two datasets as long as both have the coordinates necessary to perform the constraint and kernel operations. Note also that this algorithm only outlines the basic principles of the operations
15 of the code, a number of optimisations are used in the code itself.

One particular optimisation involves the use of Kd-Trees (Bentley, 1975) for the efficient comparison of distances. Our implementation is an extension of the SciPy algorithm (Jones et al., 2001), which uses the sliding midpoint rule to ensure a well balanced tree (Maneewongvatana and Mount, 1999), to enable comparison of haversine distances. This provides a significant performance improvement on the naive point by point comparison (which is shown in Algorithm 3, for simplicity).



---

**Algorithm 3** The 'box' collocation of an ungridded dataset onto a sampling set of coordinates $s_k$ of $K$ points. By definition the output $o$ is defined on the same spatio-temporal sampling as $s$. The distance metrics $\mathcal{D}$ are defined for each coordinate and compared with the user defined maximum $\mathbf{a}$ (the edge of the 'box').

---

INITIALIZE $o = \emptyset$

**for** k=1,...,K **do**

$\quad Q = \{u_j \forall j : \mathcal{D}(s_k, r_j) < \mathbf{a}\}$

$\quad o_k = \mathcal{K}(Q)$

**end for**

---

### 4.3.3 Ungridded to gridded

For ungridded data points which are mapped onto a gridded sample there are two options available. Either the ungridded data points can be 'binned' into the bounds defined by each cell of the sample grid using the 'bin' option, or the points can be constrained to an arbitrary area centred on the gridded sample point using the box option as described above. Either way, the
moments kernel is used by default to return the number of points in each bin or box, the mean of their values and the standard deviation in the mean.

Algorithm 4 describes this process in more detail. As with Algorithm 3, we show here the operations performed, but not the exact code-path. In reality a number of optimisations are made to ensure efficient calculations.

---

**Algorithm 4** The 'bin' collocation of an ungridded dataset onto a gridded sample set of $M$ multi-dimensional cells, as defined by the input file. Upper and lower bounds for each of the cells of the dataset ($\mathbf{b}$ and $\mathbf{a}$ respectively) are automatically deduced if not present in the data.

---

INITIALIZE $O = \emptyset$

**for** m=1,...,M **do**

$\quad Q = \{u_j \forall j : \mathbf{a}_m <= \mathbf{r}_j < \mathbf{b}_m \}$

$\quad O_m = \mathcal{K}(Q)$

**end for**

---

### 4.3.4 Gridded to ungridded

When mapping gridded data onto ungridded sample points the options available are for the nearest neighbour value, or a linearly interpolated value.

The methods used to perform the interpolation are provided by the Iris package (see http://scitools.org.uk/iris/docs/latest/iris/iris/cube.html#iris.cube.Cube.interpolate for more details), although CIS provides extended functionality by allowing users to interpolate over hybrid pressure or altitude coordinates as needed. CIS first uses Iris to interpolate in all coordinates other
than hybrid altitude or pressure, then a linear interpolation in the hybrid coordinate is performed (see Fig. 5).



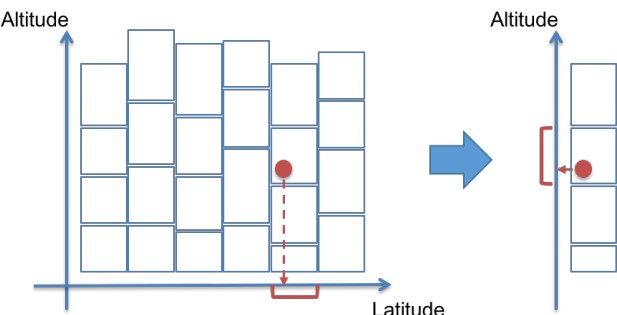

**Figure 5.** This schematic shows the collocation of gridded data onto an ungridded sampling where the altitude component of the data is defined on a hybrid height grid. CIS will first collocate the data in the coordinate dimensions (latitude, longitude, etc), to extract a single altitude column, and then perform a second interpolation on the altitude coordinate.

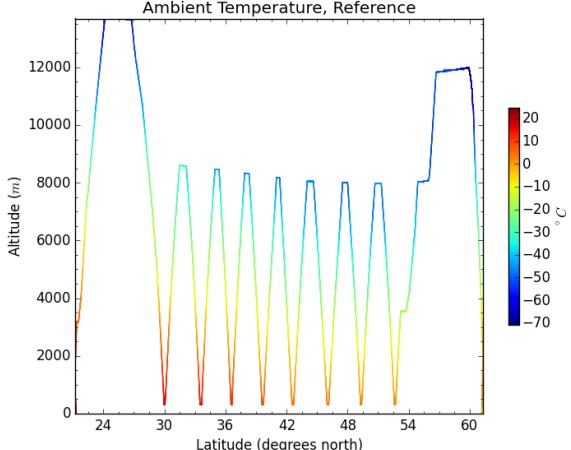

**Figure 6.** An example scatter plot from a particular aircraft measurement of ambient temperature as a function of latitude (x-axis) and altitude (y-axis) produced directly by CIS. Note the representation of temperature as the colour of the scatter points.

## 4.4 Plotting

CIS also includes a comprehensive set of plotting capabilities, allowing the analysis and comparison of the whole variety of data which can be read. This includes plots of aircraft flight tracks (see e.g. Fig. 6) and satellite imagers (Fig. 7). It also allows the plotting of heatmaps for gridded data as shown in the annual averages of AOT (Aerosol Optical Thickness) plotted in Fig. 9.

5    It also allows more detailed analysis of combined datasets, for example by plotting collocated variables against each other as a scatter plot, and even as a 2-D histogram – for highlighting the distributions when there are many thousands of points, such as in Fig. 8.



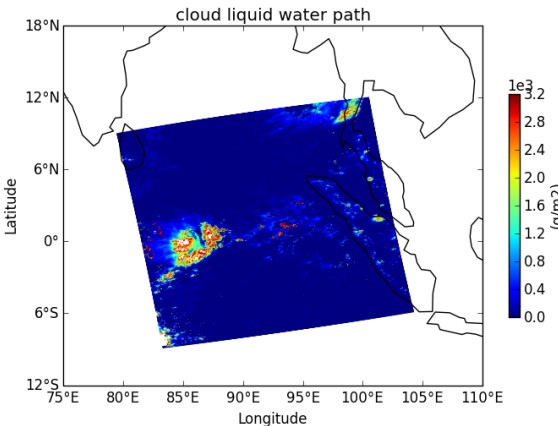

**Figure 7.** An example plot showing the cloud liquid path over the Indian Ocean just off Malaysia, retrieved by the ESA Cloud CCI product MODIS Aqua (Hollmann et al., 2013).

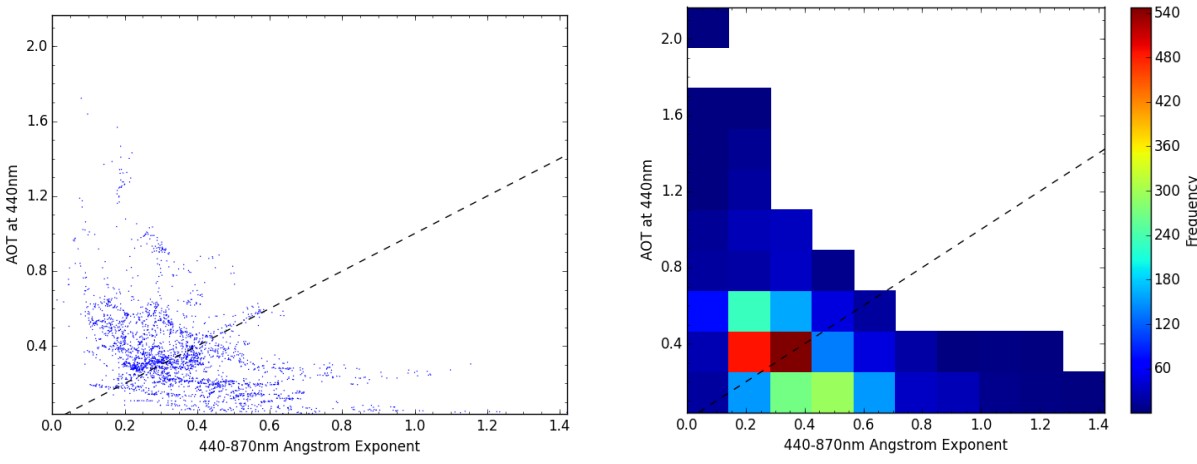

**Figure 8.** An example of plotting two collocated variables against one another as a scatter plot, and also as a 2-D histogram. This can be useful for inspecting dense scatter plots.

The plotting output is highly customisable, with more than 35 different options available for specifying everything from the axes labels, to the colour of the coastlines. The user is also able to output the plots directly to screen, for interactive visualisation, including zooming and panning, or straight to image file (including .png, .jpg, .eps, or .pdf) for publication ready plots. A full description of the plotting syntax and available options is provided in the user manual (http://cis.readthedocs.org/en/latest/plotting.html).





## 4.5 Analysis

In addition to standard analysis options as described above, CIS allows general arithmetic operations to be performed between different variables using the 'eval' command. The two variables must be on the same spatio-temporal sampling, CIS will check the data has the same dimensions but not that the points correspond to the same sampling. There are limitless possibilities, but it enables, for example, the calculation of the difference between two collocated variables as demonstrated in Fig. 9. It also allows users to easily create a mask for a dataset, for example masking all of the mean output values from a collocation for which fewer than 5 points were used.

The basic structure of the eval command is as follows:

```
$ cis eval <datagroup>... <expr> <units>
```

Where 'datagroup' has already been described above (but variables can optionally take an alias to simplify the expression), 'expr' is the expression to evaluate, and 'units' is a string describing the units which should be assigned to the new variable (and must be CF-compliant). Note that, it is actually possible to evaluate any python expression with this command, including using the numpy library, but that for security many built-in modules are unavailable.

This flexibility allows for some quite complex analysis. For example, consider the case of calculating the Ångström Exponent ($\alpha$) for AOT ($\tau_\lambda$) as measured by AERONET (AErosol RObotic NETwork) (http://aeronet.gsfc.nasa.gov/) at $870\,\text{nm}$ and $440\,\text{nm}$, where $\alpha$ is given by:

$$\alpha = -\frac{\log \frac{\tau_{\lambda_1}}{\tau_{\lambda_2}}}{\log \frac{\lambda_1}{\lambda_2}}. \tag{2}$$

This can be straightforwardly calculated using the following CIS command:

```
$ cis eval AOT_440,AOT_870:agoufou.lev20 "(-1)*(numpy.log(AOT_870/AOT_440)/numpy.
    log(870./440.))" 1 -o alpha
```

Note that we have used the numpy library to calculate the log of each of the variable arrays, and have used the (AOT) variable names in the file for $\tau$. The resulting $\alpha$ is then output to a CF compliant NetCDF4 file for further analysis or processing.

## 4.6 Statistics

Users are also able to perform a basic statistical analysis on two variables using the 'stats' command. This command has a very basic structure:

```
$ cis stats <datagroup>...
```

More than one datagroup may be specified, but the total number of variables declared in all the datagroups must be exactly two. Again, they must both be on the same spatio-temporal sampling.





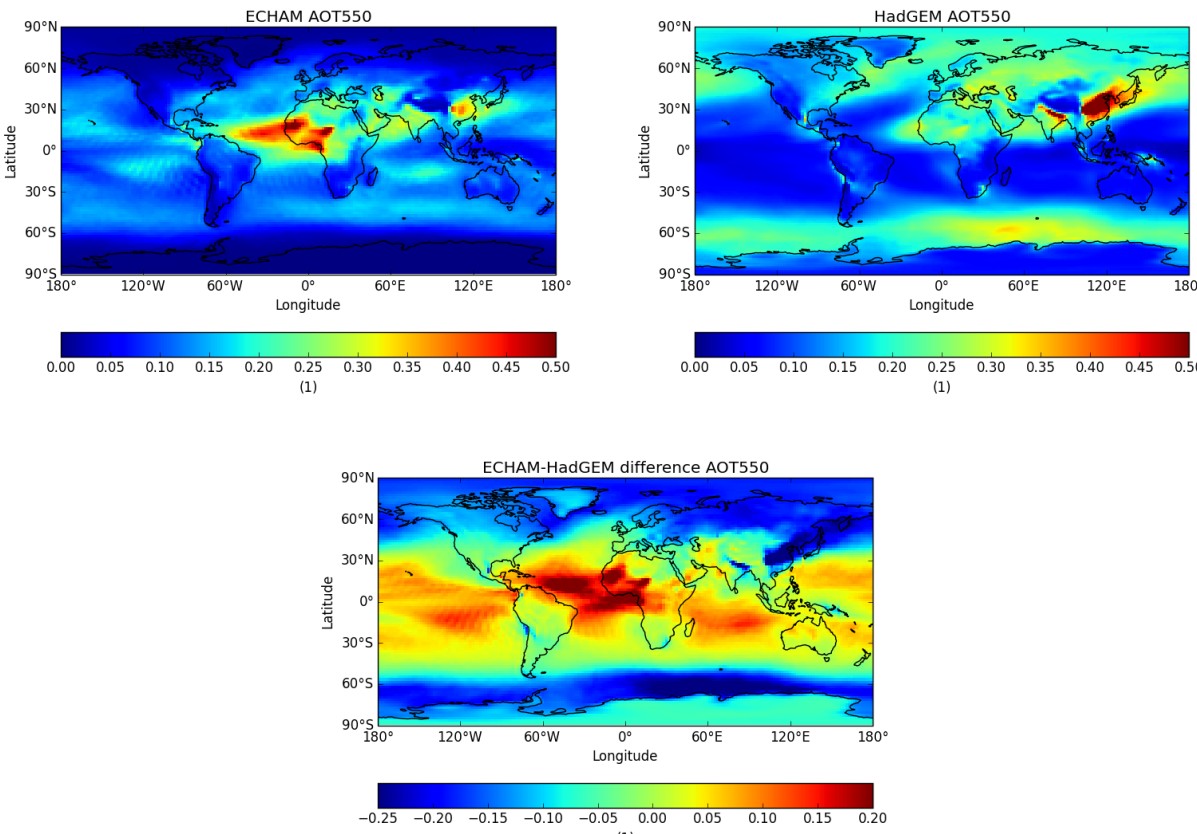

**Figure 9.** A comparison of annual average AOT at 550 nm between ECHAM and HadGEM3 across the globe.

For example, the user might wish to examine the correlation between a model data variable and actual measurements, or (as in the Ångström Exponent example above) the correlation between a calculated and measured variable. The 'stats' command will calculate:

1. Number of data points used in the analysis.

2. The mean and standard deviation of each dataset (separately).

3. The mean and standard deviation of the absolute difference ($v_2 - v_1$).

4. The mean and standard deviation of the relative difference ($(v_2 - v_1)/v_1$).

5. The Linear Pearson correlation coefficient.

6. The Spearman Rank correlation coefficient.





7. The coefficients of linear regression (i.e. $v_2 = av_1 + b$ ), $r$-value, and standard error of the estimate.

Many of these values are calculated using the SciPy library (van der Walt et al., 2011). The values are displayed on screen and can optionally be saved to a NetCDF4 file.

### 4.7 CIS as a Python library

CIS was primarily designed as a command line tool, however it is also straightforward to use some of the power of CIS in other Python modules or scripts. In particular CIS provides an interface for reading any of the datasets which CIS supports (either built-in or through user supplied plug-ins). The data is returned in a well documented data structure which provides straightforward access to the raw data, the coordinates and all associated meta-data.

Further, because the data structure returned by these routines are built on Numpy arrays, it is trivial to build these into existing Python based data analysis routines. There is also an option to return the data as a Pandas (http://pandas.pydata.org) dataframe. Pandas is an open-source data analysis package providing, amongst other things, in-depth and easy to use time-series analysis.

## 5 Example scientific workflow

Consider the comparison of a set of AERONET data with model AOT data over a given time period, for example, in order to help inform and constrain the approximations and assumptions used in the model. For the sake of this example we use ECHAM6-HAM2 (first described by Stier et al. (2005)), but as no scientific interpretation of the comparison will be sought or offered, the details of the setup are not important.

As a first step it is often useful to inspect the contents of a datafile to determine which variables it contains. This is straightforward using the 'info' command:

```
$ cis info 920801_091128_Agoufou.lev20
```

This will return a list of the variables in the file, exactly as they should be passed to other commands. The '-v' flag can be used to get more detailed information about any specific variables. Next, we might plot each of the datasets in order to examine their spatio-temporal extents, and get a feel for the magnitudes of the AOT. We can plot the AERONET data with the following command:

```
$ cis plot AOT_675:920801_091128_Agoufou.lev20
```

An example output plot is shown in Fig. 10. Note that the model data can't yet be plotted using CIS as it is extended in latitude, longitude and time. It could however be subsetted, or aggregated to reduce the dimensionality before plotting.

Next we might decide to subset the AERONET data to cover the same temporal range as the model data (which is for 2007):

```
$ cis subset AOT_675:920801_091128_Agoufou.lev20 -t=[2007]
```

In order to quantitatively compare the values we need to bring the model data onto the AERONET spatio-temporal sampling, this is straightforward using the collocation command:





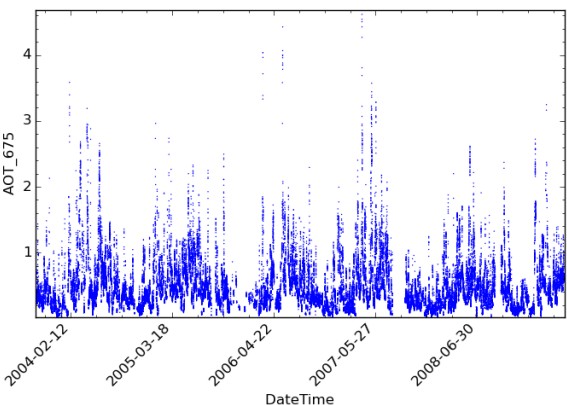

**Figure 10.** CIS plot with default options for AOT observed from a single AERONET station.

```
$ cis col TAU_2D_670nm:ECHAMHAM_AOT550_670.nc 920801_091128_Agoufou.lev20 -o
    echam_on_agoufou.nc
```

Note that this will find the nearest neighbour model data values in both space and time by default, though we could have chosen to use a linear interpolation algorithm instead. Once we have two collocated datasets we can calculate the point-wise

5   difference between the observations and the collocated model data using:

```
$ cis eval TAU_2D_670nm:echam_on_agoufou.nc AOT_675:920801_091128_Agoufou.lev20 "
    TAU_2D_670nm␣-␣AOT_675" -o echam_aeronet_agoufou_diff.nc
```

We can also use the built in analysis routines to give us an overview of the correlations between the two datasets using the `stats` command:

10   `$ cis stats TAU_2D_670nm:echam_on_agoufou.nc AOT_675:920801_091128_Agoufou.lev20`

And finally, in order to ensure a robust statistical comparison we can then aggregate the collocated data in time to provide a yearly average:

`$ cis aggregate TAU_2D_670nm:echam_on_agoufou.nc t -o echam_on_agoufou_2007.nc`

This provides the average difference of the *collocated* data values. Furthermore, because these are straightforward command-

15   line commands, we can easily script this process and repeat it for multiple AERONET stations to produce a plot of the annual difference across the globe, as shown in Fig. 11.



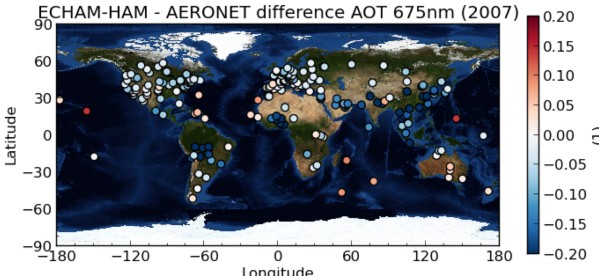

**Figure 11.** The difference between the annual average AOT measured at AERONET stations around the world and ECHAM6-HAM2 modelled values. This plot only demonstrates a type of analysis which is easy to perform with CIS, no scientific critique of these differences is offered.

## 6   Conclusions and summary

The intercomparison of observational and model data is a crucial aspect of modern climate science. There exist a few tools to work with gridded NetCDF datasets, but very few in support of process studies using assorted data sources, and none which allow generic intercomparison of multiple ungridded and/or gridded datasets.

5   Here we have demonstrated the power and use of CIS – a new universal tool for the inter-comparison of model, remote sensing and in-situ climate data. The open and extensible nature of the tool allows for the easy and reproducible collocation, aggregation, subsetting and analysis of a huge variety of data sources on everything from laptops to large processing clusters. Further, the ability to extend the data sources compatible with CIS through user developed plugins provides the opportunity for a shared tool to serve a diverse community.

10   Further development of CIS is ongoing and we hope to include a number of new features in the future, such as an extended Python Application Programming Interface (API), hybrid gridded/ungridded data structures and improved time-series analysis. However, the growth of a user community (centred around our website) will help decide on the priority and best implementation of such features through user feedback and users actively engaging in development. All descriptions of functionality are correct as of version 1.3.2 (Watson-Parris et al., 2016), future releases and announcements can be found on the CIS website: http:

15   //cistools.net.

## 7   Code availability

The CIS source code is available on GitHub at https://github.com/cedadev/cis, and the binary is available for easy installation on Windows, OS X and Linux through conda using the 'cistools' channel (see http://cistools.net/get-started#installation for more details). CIS is also pre-installed on the UK JASMIN analysis platform (JASMIN runs a number of Red hat Enterprise

20   Linux 6 scientific computing virtual machines for which CIS is pre-installed. See http://www.jasmin.ac.uk for more details). Detailed documentation and help pages can be found at http://cis.readthedocs.org.



CIS is a tool for working wth a wide variety of data, however none of the datasets used or described within this paper are supplied with the tool and should be obtained directly through their respective providers.

*Acknowledgements.* We would like to acknowledge the guidance and support of Stephen Pascoe through his role in CEDA during the first phases of development, and Caroline Poulsen (Remote Sensing Group, EOAS Division, RAL Space) who provided invaluable user feedback.

5   Each development phase has been supported by STFC JASMIN capital grants, and a NERC Big Data call.



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

## Appendix A:  Table of definitions

**Table 5.** A table of terms in this paper

| Term | Definition |
|---|---|
| Aggregate | The process taking ungridded data and performing averaging over time and/or space to produce a gridded output. |
| Point-wise operation | An operation carried out on each data point individually, usually on ungridded data. |
| Gridded | Any regularly gridded data-set for which points can be indexed using $(i, j, k, ...)$ where $i$, $j$ and $k$ are integers. |
| Ungridded | Any data which is not regularly gridded, in general it is assumed each $(x, y, z)$ point is independent of every other point. |

## Appendix B:  Plugin development

In this section we describe two specific ways that users are able to easily extend the functionality provided by CIS. The plugins are short pieces of Python code that users can write themselves and which CIS will then automatically incorporate. Our website offers functionality for users to upload new plugins to be shared with the wider CIS community.



A detailed description of the development of CIS plugins, and a number of increasingly in-depth tutorials can be found in the CIS documentation (http://cis.readthedocs.org/en/stable/plugin_development.html), here we provide only an overview of the basic plugin structures.

### B1  Data plugins

CIS uses the notion of a 'data product' to encapsulate the information about different types of data. Users can write their own products for reading in different types of data – referred to as 'plugins'. These products (or plugins, if provided by the user) are concerned with interpreting the raw data and its coordinates and producing a single self describing data object conforming to the CommonData interface (see Fig. 1). They follow a defined structure so that they can be automatically included and used by the tool. We briefly describe that structure here.

All plugins must subclass the `AProduct` abstract class (this class defines the structure described here, and indicates to CIS the type of plugin the user has supplied), and are therefore forced to provide an implementation for the following methods:

  – `get_file_signature(self)` This method returns a list of regular expressions to match the product's file naming convention. CIS will use this to decide which data product to use for a given file. The first product with a signature that matches the filename will be used.

– `create_coords(self, filenames)` Return the coordinates from one or more files. Note that this method may have to make certain assumptions about the file in order to return a single coordinate set.

  – `create_data_object(self, filenames, variable)` Creates and returns a CommonData object for a given variable from a list of filenames.

The underlying I/O layers are also available for the plugins to use (such as NetCDF reading) which ensures the writing of
plugins is as straightforward as possible.

### B2  Collocation

Users can also write their own plugins for performing the collocation of two datasets. There are three main objects used in the collocation which the user is free to override: The collocator; the constraint; and the kernel. The basic design is that the collocator loops over each of the sample points, calls the relevant constraint to reduce the number of data points, and then calls
the kernel which returns a single value for the collocator to store.

The main plugin which is available is the collocation method itself. A new one can be created by subclassing `Collocator` and providing an implementation for the main 'collocate' method. This method takes a number of points and applies the given constraint and kernel methods on the data for each of those points. It is responsible for returning the new data object to be written to the output file.

The constraint object limits the data points for a given sample point somehow. The user can also add a new constraint method by subclassing `Constraint` and providing an implementation for the method `constrain_points`. The final plugin type



is the `Kernel` which is used to convert the constrained points into values in the output, many examples of which are listed in Table 4.

Although we provide an outline here please see the technical documentation for more details (http://cis.readthedocs.org/en/latest/analysis_plugin_development.html).