# Peer review of "Community Intercomparison Suite (CIS) v1.3.2: A tool for intercomparing models and observations"

_Geoscientific Model Development, 2016_

## Referee Comment (RC1) · Anonymous Referee #1 · 19 Apr 2016

This paper presents the Community Intercomparison Suite, a freely-available Python package for processing, analysing and plotting model and observational data of various type (in-situ and satellite). This software represents a valuable tool for data analysis in the climate community. It is well documented in this paper and a detailed documentation is also available on the CIS wepage.

The manuscript is well written and mostly clear, although some minor improvements are required before publication in GMD. See some suggestions in the following.

Page 1, Line 17: please add a reference for the statement about CMIP6.

Page 2, Line 17: there are other tools which are able to read and process observational

data, for example the ESMValTool (Eyring et al., GMDD 2015) and the PCMDI metrics package (Gleckler et al., EOS 2015). Please mention them in the text.

Page 2, Line 23: I would be more specific here, by writing: "e.g., from the obs4MIPS project, Texeiera et al., 2014".

Page 7, Line 24: what if the datafile does not contain units information? This is often the case when working with ungridded data which do not comply any standard. Does CIS attempt to correct units inconsistency/errors in the input data? Does the user have any control on that?

Page 11, Line 15: how are missing values in non-NetCDF files handled (i.e., if there is no explicit _FillValue attribute)?

Page 12, Line 12: are other regridding methods (e.g. area/energy-conserving) available? Is a support for non-regular grids (e.g., ocean grids) planned for the future?

Page 17, Line 13: this sentence is unclear. Please clarify what are the security issues mentioned here.

Page 21, Line 4: as mentioned above, there are other tools which allow to compare multiple dataset. Please rephrase this sentence.

Appendix A: the Table of definitions is quite short (only 4 terms). I would suggest moving the definitions to their first occurrence in the main text (as a footnote or similar).

Page 24, Line 22: are the user-provided plugins tested before they are made available to the CIS community? Please specify.

TEXT CORRECTIONS

Page 3, Line 16: *it's* → *its*.

Page 8, Algorithm 1: *iff* → *if*.

Page 9, L26: *an collocation* → *a collocation*.

---

## Referee Comment (RC2) · Anonymous Referee #2 · 22 Jul 2016

This study describes the development and use of the Community Intercomparison Suite (CIS). CIS represents a novel and significant model development, allowing intercomparison of diverse climate model output and observations in both gridded and ungridded formats. In general the paper is well written with methods described clearly and thoroughly. In particular section 5 (Example scientific workflow) stands out as a very useful 'how to' guide to using the CIS.

Given the above I therefore recommend publication after the authors address the (predominantly) minor points outline below:

Page 5 Line 25: 'These arrays make take on any shape, though they must all be the same' . This line took me some time to understand. I presume what the authors mean

is that the structure of the arrays can be any shape but all the data values (lon,lat etc) need to conform to that structure? Please could the authors clarify and amend. I would also reference Figure 2b here.

Formula 1 (page 6): Please could the authors explain this formula more fully. I understand the inequality but what is x 5 y (Likewise in algorithm 1). I understand the need to adopt a consistent notation but this notation needs to be more clearly described (particularly if its not standard vector or set but a mixture)

Page 6 Line 4: change 'Øto' to 'Ø to'

Algorithm 1 (page 8) change 'iff 'to 'if'

Page 8 Line 13: change font on '5°'

Page 11 Line 17: Could the authors define h_sep and a_sep to aid the reader in understanding this example.

Table 4 (page 12): Equations for the arithmetic mean and standard deviation are not formatted correctly.

Page 12 Lines 17-18 to Page 13 Lines 1-2: '….., those dimensions are ignored for the purposes of the collocation, and will be present in the output' Could the authors please clarify this (important) point perhaps using a example. It's my understanding that if (for example) you want to collocate a 2D array (data) onto a 3D sample array you will end up with a 2D result but if you collocate a 3D data array onto a 2D sample array you get a 3D array (i.e. your resulting array will always have the same dimensions as your data array). Could the authors also discuss the consequences of mis-matching the sample and data array dimensions.

Page 17 Line 27-28: ', but the total number of variables declared in all the datagroups must be exactly two'. The number of variables is integer so 'exactly two' is redundant.

Page 19 Line 13: Based on the filename, the authors use AERONET data from

Agoufou I would suggest here referring to 'AERONET data from the Agoufou station' to aid readers unfamiliar with the AERONET stations.

Page 19 Line 25: 'Note that the model data can't be plotted....' Throughout the paper you note when the CIS commands will not work. It would be useful to would-be users to include likely error messages when these mistakes are made.

Page 20 Line 10: Please could the authors add some description of what the results look like when you use the stats command.

Figure 11 (Page 21): Could the authors also include the commands to plot this map. This last step (plotting) seems to be missing from section 5. If possible it would also be useful to add a few more examples of the plotting capabilities of the system.

Appendix A: I feel that table 5 could be expanded to include (among others), collocation. I would also suggest adding a table defining the common inputs in CIS commands (i.e. datagroup, samplegroup etc )

---

## Author Comment (AC1) · 11 Aug 2016

This paper presents the Community Intercomparison Suite, a freely-available Python package for processing, analysing and plotting model and observational data of various type (in-situ and satellite). This software represents a valuable tool for data analysis in the climate community. It is well documented in this paper and a detailed documentation is also available on the CIS wepage.

The manuscript is well written and mostly clear, although some minor improvements are required before publication in GMD. See some suggestions in the following.

Page 1, Line 17: please add a reference for the statement about CMIP6.

Page 2, Line 17: there are other tools which are able to read and process observational

data, for example the ESMValTool (Eyring et al., GMDD 2015) and the PCMDI metrics package (Gleckler et al., EOS 2015). Please mention them in the text.

Page 2, Line 23: I would be more specific here, by writing: "e.g., from the obs4MIPS project, Texeiera et al., 2014".

Page 7, Line 24: what if the datafile does not contain units information? This is often the case when working with ungridded data which do not comply any standard. Does CIS attempt to correct units inconsistency/errors in the input data? Does the user have any control on that?

Page 11, Line 15: how are missing values in non-NetCDF files handled (i.e., if there is no explicit _FillValue attribute)?

Page 12, Line 12: are other regridding methods (e.g. area/energy-conserving) available? Is a support for non-regular grids (e.g., ocean grids) planned for the future?

Page 17, Line 13: this sentence is unclear. Please clarify what are the security issues mentioned here.

Page 21, Line 4: as mentioned above, there are other tools which allow to compare multiple dataset. Please rephrase this sentence.

Appendix A: the Table of definitions is quite short (only 4 terms). I would suggest moving the definitions to their first occurrence in the main text (as a footnote or similar).

Page 24, Line 22: are the user-provided plugins tested before they are made available to the CIS community? Please specify.

TEXT CORRECTIONS

Page 3, Line 16: *it's → its*.

Page 8, Algorithm 1: *iff → if*.

Page 9, L26: *an collocation → a collocation*.

**Response to Anonymous Referee #1**

Thank you for your helpful feedback. We have implemented each of the suggestions you mentioned with the exception of a couple which are responded to individually below:

Page 11, Line 15: This good question is now dealt with in the Core Functionality section.

Appendix A: We feel it's useful to have a summary of definitions as these terms are used throughout the paper in quite specific ways. We have also added 'collocate' as suggested by Referee 2.

We have also made a few small changes unrelated to the reviewer comments to reflect the latest version of CIS available (1.4.0) and expand on the acknowledgments.

Kind Regards

[revised manuscript text omitted]

---

## Author Comment (AC2) · 11 Aug 2016

This study describes the development and use of the Community Intercomparison Suite (CIS). CIS represents a novel and significant model development, allowing intercomparison of diverse climate model output and observations in both gridded and ungridded formats. In general the paper is well written with methods described clearly and thoroughly. In particular section 5 (Example scientific workflow) stands out as a very useful 'how to' guide to using the CIS.

Given the above I therefore recommend publication after the authors address the (predominantly) minor points outline below:

Page 5 Line 25: 'These arrays make take on any shape, though they must all be the same' . This line took me some time to understand. I presume what the authors mean

is that the structure of the arrays can be any shape but all the data values (lon,lat etc) need to conform to that structure? Please could the authors clarify and amend. I would also reference Figure 2b here.

Formula 1 (page 6): Please could the authors explain this formula more fully. I understand the inequality but what is x 5 y (Likewise in algorithm 1). I understand the need to adopt a consistent notation but this notation needs to be more clearly described (particularly if its not standard vector or set but a mixture)

Page 6 Line 4: change 'Øto' to 'Ø to'

Algorithm 1 (page 8) change 'iff 'to 'if'

Page 8 Line 13: change font on '5°'

Page 11 Line 17: Could the authors define h_sep and a_sep to aid the reader in understanding this example.

Table 4 (page 12): Equations for the arithmetic mean and standard deviation are not formatted correctly.

Page 12 Lines 17-18 to Page 13 Lines 1-2: '….., those dimensions are ignored for the purposes of the collocation, and will be present in the output' Could the authors please clarify this (important) point perhaps using a example. It's my understanding that if (for example) you want to collocate a 2D array (data) onto a 3D sample array you will end up with a 2D result but if you collocate a 3D data array onto a 2D sample array you get a 3D array (i.e. your resulting array will always have the same dimensions as your data array). Could the authors also discuss the consequences of mis-matching the sample and data array dimensions.

Page 17 Line 27-28: ', but the total number of variables declared in all the datagroups must be exactly two'. The number of variables is integer so 'exactly two' is redundant.

Page 19 Line 13: Based on the filename, the authors use AERONET data from

Agoufou I would suggest here referring to 'AERONET data from the Agoufou station' to aid readers unfamiliar with the AERONET stations.

Page 19 Line 25: 'Note that the model data can't be plotted. . ..' Throughout the paper you note when the CIS commands will not work. It would be useful to would-be users to include likely error messages when these mistakes are made.

Page 20 Line 10: Please could the authors add some description of what the results look like when you use the stats command.

Figure 11 (Page 21): Could the authors also include the commands to plot this map. This last step (plotting) seems to be missing from section 5. If possible it would also be useful to add a few more examples of the plotting capabilities of the system.

Appendix A: I feel that table 5 could be expanded to include (among others), collocation. I would also suggest adding a table defining the common inputs in CIS commands (i.e. datagroup, samplegroup etc )
* * *
**Response to Anonymous Referee #2**

Thank you for your helpful feedback. We have implemented each of the suggestions you mentioned with the exception of a few which are responded to individually below:

Formula 1 (page 6): We have reworded this formula to hopefully be clearer

Page 6 Line 4, Algorithm 1 (page 8), Page 8 Line 13 and Table 4 (page 12): These are formatting errors which we can't reproduce – perhaps it is a problem with the particular PDF viewer being used?

Appendix A: We have extended the table of definitions to include collocation, though we feel that a table of the common CIS inputs best belongs in the online user manual.

We have also made a few small changes unrelated to the reviewer comments to reflect the latest version of CIS available (1.4.0) and expand on the acknowledgments.

Kind Regards

[revised manuscript text omitted]

---

## Author Response (AR2)

Dear Fiona,

Many thanks for taking the time to look the paper over again, I've responded to each of your points in turn below and included a copy of the manuscript highlighting the changes.

Kind regards,

Duncan

Dear Duncan and co-authors,

Many thanks for uploading a revised version of your manuscript. Having read all the reviewer comments and your responses to them, I note the following:

1. In relation to Reviewer #1, I feel that you have adequately addressed the main comments. In the case of Appendix A, a table is more suitable than adding definitions as footnotes in the main manuscript and I support your retention of Table 5 in Appendix A. However, I felt that you misinterpreted this Reviewer's comments in relation to Obs4MIPs. Firstly, Obs4MIPs is not the sole source of aggregated observational data and Reviewer #1 suggested that you include its name as an example rather than just referring to Teixeira et al. It is only a minor point but could I please ask that you change the statement "Aggregated observational data is often available from the obs4MIPS project (Teixeira et al., 2014)". to "Aggregated observational data is often available (e.g. from the Obs4MIPs project, Teixeira et al., 2014)".

> Thank you for pointing that out, I've updated that sentence as suggested.

2. In relation to Reviewer #2, there were a number of comments made by this Reviewer which did not appear to have been addressed. The first was in relation to the explanation of Formula 1. Although you indicated that it had been re-worded, I saw no evidence of that. Are you sure that you uploaded the most recent revised manuscript?

> The extended explanation is now included, apologies.

Secondly, this reviewer also requested that some indication of the error messages that CIS would give in the case where a would-be user was trying to do something not allowed e.g. Note that model data can't yet be plotted (Section 5). Could you please add such messages concisely in order to address the Reviewer's comments but without making the manuscript too unwieldly?

These are now included where appropriate.

Finally, the Reviewer requested that you add a discussion about mismatching the sample and data array dimensions. I note that you've added a new figure to explain this but could you add a couple of lines in response to this comment? Otherwise, you addressed all other comments sufficiently.

I assumed this to be a restating of the original question and that the figure would be sufficient as there is already a paragraph of text devoted to that discussion. Perhaps I've misunderstood though?

The only other comments that I had were minor as follows:

3. Update Eyring et al. 2016a and 2016b references
    Fixed, apologies

4. Give full name for PMP (Pg 2, line 18)
    Done

5. Correct spelling (Therefore -> Therefore) on pg 13, line 12.
    Fixed

6. Caption of (newly added) Figure 4: Change "an gridded sampling" to "a gridded sample"
    Fixed

Could I please ask you to respond to these comments and once completed (and uploaded), I will recommend that your manuscript be accepted for publication in GMD.

Thanks again,
Fiona O'Connor

[revised manuscript text omitted]
 (otherwise CIS will return an error message: "Stats command requires exactly two variables (n were given)"). Again, they

[Figure]

**Figure 10.** A comparison of annual average AOT at $550\,\mathrm{nm}$ between ECHAM and HadGEM3 across the globe.

must both be on the same spatio-temporal sampling (otherwise CIS will return an error message that "operands could not be broadcast together").

For example, the user might wish to examine the correlation between a model data variable and actual measurements, or (as in the Ångström Exponent example above) the correlation between a calculated and measured variable. The 'stats' command will calculate:

1. Number of data points used in the analysis.

2. The mean and standard deviation of each dataset (separately).

3. The mean and standard deviation of the absolute difference ($v_2 - v_1$).

4. The mean and standard deviation of the relative difference ($(v_2 - v_1)/v_1$).

5. The Linear Pearson correlation coefficient.

6. The Spearman Rank correlation coefficient.

7. The coefficients of linear regression (i.e. $v_2 = av_1 + b$ ), $r$-value, and standard error of the estimate.

Many of these values are calculated using the SciPy library (Jones et al., 2001). The values are displayed on screen and can optionally be saved to a NetCDF4 file.

**4.7 CIS as a Python library**

CIS was primarily designed as a command line tool, however it is also straightforward to use some of the power of CIS in other Python modules or scripts. In particular CIS provides an interface for reading any of the datasets which CIS supports (either built-in or through user supplied plug-ins). The data is returned in a well documented data structure which provides straightforward access to the raw data, the coordinates and all associated meta-data.

Further, because the data structure returned by these routines are built on Numpy arrays, it is trivial to build these into existing Python based data analysis routines. There is also an option to return the data as a Pandas (http://pandas.pydata.org) dataframe. Pandas is an open-source data analysis package providing, amongst other things, in-depth and easy to use time-series analysis.

Version 2.0 of CIS is planned to include full support for all of the main CIS commands through the Python interface. For an outline of our future plans for CIS please see www.cistools.net/roadmap.

**5 Example scientific workflow**

Consider the comparison of a set of AERONET data from the Agoufou station with model AOT data over a given time period, for example, in order to help inform and constrain the approximations and assumptions used in the model. For the sake of this example we use ECHAM6-HAM2 (first described by Stier et al. (2005)), but as no scientific interpretation of the comparison will be sought or offered, the details of the setup are not important.

As a first step it is often useful to inspect the contents of a datafile to determine which variables it contains. This is straightforward using the 'info' command:

```
$ cis info 920801_091128_Agoufou.lev20
```

This will return a list of the variables in the file, exactly as they should be passed to other commands.  Variables can also be specified in the usual way to get more detailed information about any specific variables. Next, we might plot each of the datasets in order to examine their spatio-temporal extents, and get a feel for the magnitudes of the AOT. We can plot the AERONET data with the following command:

```
$ cis plot AOT_675:920801_091128_Agoufou.lev20
```

An example output plot is shown in Fig. 11. Note that the model data can't yet be plotted using CIS as it is extended in latitude, longitude and time (if we tried CIS would return an error message telling us "Data is not 1D or 2D - can't plot it on a map."). 
[revised manuscript text omitted]